# Uptake of complete postnatal care services and its determinants among rural women in Southern Ethiopia: Community-based cross-sectional study based on the current WHO recommendation

Aklilu Habte[1]*, Feleke Gebiremeskel[2], Misgun Shewangizaw[3], Samuel Dessu[4], Mustefa Glagn[2]

1 Department of Reproductive Health, School of Public Health, College of Medicine and Health Sciences, Wachemo University, Hosanna, Ethiopia, 2 School of Public Health, College of Medicine and Health Sciences, Arba Minch University, Arba Minch, Ethiopia, 3 Health Education and Promotion Unit, School of Public Health, College of Medicine and Health Sciences, Arba Minch University, Arba Minch, Southern Ethiopia, 4 Department of Public Health, College of Medicine and Health Sciences, Wolkite University, Wolkite, Southern Ethiopia

* akliluhabte57@gmail.com

## Abstract

### Background

Postnatal care services are a constellation of preventive care, practices, and assessments designed to identify and manage maternal and newborn complications during the first six weeks after birth. Recognizing the role of the appropriate PNC at this critical time, the World Health Organization recommended four visits as a complete PNC for all postpartum mothers and newborns to ensure their survival. Although there have been numerous studies on the factors affecting the general PNC service in Ethiopia, there is a shortage of evidence on the use of complete postnatal care services and its determinants. Therefore, the objective of this study was to assess the uptake of full postnatal care service and its determinants among women who recently gave birth in the Ezha district of southern Ethiopia.

### Methods

A community-based cross-sectional study was conducted in Ezha district. A two-stage sampling technique was applied. A total of 568 mothers who stayed for at least 6 weeks after childbirth from selected Kebeles were included in the study using computer-generated random numbers. Data collected through pre-established, structured, and interviewer-administered questionnaires were entered into EpiData3.1 and exported to SPPS version 23 for analysis. The Principal Components Analysis (PCA) was carried out to assess the wealth status of participants. The Multivariable logistic regression model has been fitted to identify the determinants of complete post-natal care service uptake.

**Data Availability Statement:** All relevant data are within the manuscript and its Supporting Information files.

**Funding:** The author(s) received no specific funding for this work.

**Competing interests:** The authors have declared that no competing interests exist.

**Abbreviations:** ANC, Antenatal Care; AOR, Adjusted Odds Ratio; HEWs, Health Extension Workers; MMR, Maternal Mortality Ratio; PCA, Principal Component Analysis; PNC, Postnatal Care; PPV, Postpartum Visits; SDG, Sustainable Development Goal; SPSS, Statistical product and service solutions; WHO, World Health Organization.

## Results

The overall uptake of complete postnatal care services in the study area was 23.9% [95% CI: (19.9, 27.5)]. The factors namely; maternal education of secondary and above [AOR: 4.31, 95%CI: 2.15, 8.05], having four and more antenatal visits [AOR: 4.03, 95%CI: 1.83, 8.85], Caesarean delivery [AOR: 3.75, 95%CI: 1.78, 7.92], having good knowledge on PNC [AOR: 4.31, 95%CI: 2.34, 9.04], and being a model household [AOR: 3.61, 95%CI: 1.97, 6.64] were recognized as determinants complete postnatal care uptake.

## Conclusion and recommendation

Complete post-natal care services uptake in the study area was low. Thus, a due emphasis should be given to behavioral change communication activities to improve maternal knowledge on PNC and enhancing adequate ANC uptake by health workers in the district. Besides, health extension workers in the district need to work on creating a model household through continuing education, support, and follow-up.

## Introduction

Despite several global and national initiatives aimed at improving maternal and newborn health, death remains a global challenge [1]. The maternal mortality ratio (MMR) was about 20 times higher in low and middle-income countries (LMICs) than in high-income countries [2]. There were approximately 303,000 maternal deaths worldwide in 2015, with Sub-Saharan Africa (SSA) accounting for more than 65% (201,000) of maternal deaths. Country Ethiopia represents a maternal mortality ratio (MMR) of 353 per 100000 live births with a neonatal mortality ratio of 29 per 1000 live births [3]. According to the 2017 UNICEF report, Ethiopia was grouped among five countries accounting for 50% of global neonatal death like India, Pakistan, Nigeria, and the Democratic Republic of the Congo [4]. Despite a myriad of reasons, most of these deaths occurred at birth or during the postnatal period because of inadequate care [5, 6].

World health organizations(WHO) defined Postnatal care services as a constellation of preventive care, practices, and assessments designed to identify and manage maternal and newborn complications during the first six weeks after birth [7]. Recognizing the role of appropriate PNC uptake during this critical period, WHO recommended four PNC visits to mothers and newborns to improve their survival. The recommended post-natal visits are; in the first 24 hours, on the third day, between days 7 and 14, and 6 weeks of childbirth [7, 8]. Once a mother has been given all these visiting schedules can be considered as she is getting the complete postnatal care (CPNC) service [7, 9].

Compliance with all recommended visits promotes and sustains maternal and newborn well-being by identifying and managing birth-related complications [7, 10, 11]. Besides, post-natal contact within the first six weeks allows women to reach their breastfeeding goals and address common postnatal concerns. Therefore, all women should ideally have a minimum of four contacts with a qualified health care provider within the first 42 days of childbirth [7, 12].

To optimize the health of women and newborns, the PNC should be delivered with multiple visits, rather than a single contact, with care and support adapted to the acute postpartum problems and the individual needs of each woman and newborn [13]. Pieces of evidence suggest that receiving all these schedules of PNC visits by qualified health care providers at a 90% coverage level could prevent up to 310,000 newborn deaths per year in Africa [5, 14].

Although the use of PNC has increased over the past 20 years, there are enormous inequities in the comprehensiveness of care concerning timing and content [15, 16]. More than 90% of women in developed regions, such as the Americas and Europe, complied with WHO recommendations, compared to only 37% in low- and middle-income countries [17]. Only 13% of mothers in sub-Saharan Africa were able to achieve complete postnatal care (CPNC) in compliance with WHO recommendations [18]. While improvements have been made to increase the accessibility of most maternal and child health services, reports have shown that the national prevalence of PNC use in Ethiopia is still low at 17% coverage [19]. Studies in countries such as Myanmar, India, Ghana, Tanzania, and Ethiopia have shown that the level of use of the CPNC ranges from 10% to 60% [9, 20–23].

Over the past 20 years, several trials, mainly in Asian countries and more recently in sub-Saharan Africa, have been tried at the community level to obtain adequate PNC participation through the implementation of home visiting programs, aided case management, and referral in the event of neonatal illness [24]. All of these strategies have been implemented across multiple service delivery platforms and good progress has been made in reducing maternal and newborn deaths [24, 25]. CPNC service delivery to women following childbirth was one of the key recommendations of the Ending Preventable Maternal Mortality (EDPMM) initiative and the Sustainable Development Goals (SDGs) to reach the global goal of reducing maternal mortality (70 maternal deaths per 100,000 live births) [26, 27]. The Federal Ministry of Health of Ethiopia has implemented several high-impact interventions in recent years to enhance maternal and child health services and adequate PNC service provision is part of these interventions with a 95% coverage plan by 2020 [28].

Existing studies focus on the assessment of the general PNC of at least one visit [29–34] and little has been done on the use of CPNC [9, 21–23]. Also, the average and range of visits and contents of care during postpartum visits (PPV) by skilled health care providers was not adequately addressed. Although this study was limited to a small area, it looked at women who did not receive complete PNC services. It is important to fully understand the determinants of adopting CPNC services to assist in the development and implementation of evidence-based approaches to improve service. Therefore, the study was aimed at determining the level of CPNC uptake and its determinants among rural women in the Ezha district of southern Ethiopia, 2019. The results of this study could contribute to program planning and policy-making aimed at improving maternal and child health.

## Materials and methods

### Study setting and design

The study was conducted in the Ezha district, in the Guraghe area of southern Ethiopia, located 198 km south of Addis Ababa (the capital of Ethiopia). The district is made up of 28 rural kebeles (*Kebele*: *the smallest administrative unit in the current Ethiopian government structure under the district*). According to the 2007 CSA projections, the total population of the district for 2019 was 112,948. The predicted pregnancies and live births in the district were 3801 and 3603 respectively. Four health centers, one unprofitable NGO clinic, and 28 health posts (one in each kebele) make up the primary health care units that provide maternal and child health services. This Community-based cross-sectional study was conducted from February 10 to 10 March 10, 2019.

### Populations of the study

The Source population consisted of all women who gave birth within the past 12 months and who resided in the district and stayed there six weeks or more after delivery. Those women

from the selected kebeles in the district who met the eligibility criteria were the study population. Postpartum mothers who lived in the study area for a minimum of six months were included. Mothers with postpartum periods (PPP) of less than 6 weeks and who were seriously ill during the data collection period were excluded from the study.

## Sample size determination

Initially, the sample size for this study was calculated using the stat calc menu of Epi-info version 7. A sample size of 515 was found using the parameters of the single population ratio formula; the estimated prevalence of CPNC use of 28.4% from a study in northern Ethiopia [23], 95% confidence level, 5% degree of precision, 1.5 design effect and 10% non-response. Second, a double population formula was used based on the factors associated with the adoption of the CPNC. Among those factors selected, the largest sample size(n = 568) was obtained by considering percent of CPNC in unexposed(i.e. women who didn't attend secondary education = 16.06), AOR of 2.16 [9], and by using parameters of 80% power, 95% confidence level, 5% degree of precision, the ratio of unexposed to exposed 1, non-response rate of 10% and a design effect of 1.5. Lastly, the sample size based on two population proportions consideration (n = 568) was larger than the sample size for a single population proportion (n = 515) and the maximum one was taken as the final sample size for the study.

## Sampling techniques

A two-stage sampling technique was applied to obtain study participants. The district consists of 28 Kebeles and 10 Kebeles have been randomly selected through the lottery method. The list of women who gave birth in the past 12 months and who remained for six weeks or more after delivery was obtained from the registry of each health post, as each birth was recorded daily. Eligible women in each of the selected Kebeles were counted and reassured by health extension workers for their presence. Codes/numbers were then provided for homes with eligible study participants and a sampling frame had been developed for each kebele. The sample size for each Kebele was calculated using a proportional allocation. Finally, the study participants were selected using a computer-generated random number and interviewed at their homes. If more than one eligible woman resides in the same home, only one was chosen by lottery method. When the selected households were closed during data collection, interviewers went back three or more times at different intervals.

## Data collection tools, methods, and procedures

The data was collected by six graduate nurses through face-to-face interviews with the supervision of two public health officers working in the district. A pre-tested structured questionnaire was developed by looking at different pieces of literature in the area of interest [7, 21–23, 35, 36]. The questionnaire was designed in a way to capture information on socio-demographic, obstetric, knowledge, and health care system-related characteristics of respondents. Household's socioeconomic status was assessed by using a tool adapted from EDHS 2016, composed of multiple items like; household assets, livestock ownership, crop production in quintals, average estimated monthly income, having agricultural land in hectares, and residential home with its infrastructures [19]. Supervisors and data collectors used the guidance of the local Health Development Army (HDA) and community voluntary workers in each Kebele to accessed houses of the sampled women. The study participants were interviewed at their residence.

## Data quality management

The questionnaire used for data collection was first prepared in English and translated into the local language by an expert in that language and then translated into English to ensure consistency with the original meanings. Intensive day-long training was provided to data collectors and supervisors on data collection methods and instruments. A pretest was done for 5% of the sample size (29 women) in the Cheha district one week before commencing the actual data collection. All necessary corrections were made based on the outcome of the pretest test to avoid confusion and to better complete the questionnaires. The principal investigator and supervisor followed up daily during the entire data collection period. Each day, following data collection, supervisors, and the PI reviewed and verified the completeness of each questionnaire, and the necessary feedback was provided to the data collectors overnight.

## Definition and operationalization of variables of the study

The outcome variable of this study is the complete post-natal care (CPNC) utilization. A woman was considered to have CPNC when she received all four WHO-recommended visits within the first six weeks of their last birth; within 24 hours, day 3, 7–14 days, and to the sixth week thereafter through a home or institutional visits by qualified health care providers [7, 9]. This self-reported visit was confirmed by reviewing the postpartum enrolment registries at the health post. Lastly, the outcome variable was dichotomized to 1: YES, and 0: NO.

**Skilled health care provider.** A health professional such as a doctor, nurse, midwife, health officers, or health extension worker who works in health facilities and has been trained to provide postnatal care services [19].

**Knowledge on PNC.** This is the familiarity or awareness of respondents of having PNC information, post-natal care content, PNC advantages, and consequences of not receiving PNC, frequency, and timing of PNC, maternal and newborn danger signs during PPP. A composite knowledge indicator was constructed and classified as poor, moderate, and good knowledge when a participant answered no more than three, four to five, and no less than six of the ten knowledge assessment questions correctly, respectively [37].

**Being a model household (MHH).** Participants who implemented all health extension packages and obtained a certificate of recognition and appreciation from the relevant organizations [38].

**Autonomy to maternity care.** This is how resources are identified and controlled when women should seek maternal health services and classified as: autonomous, if she decides alone or with her husband (jointly) to seek maternal and child health care; otherwise not autonomous, it means a husband alone or a third party decided on the use of the services [19].

**Wontedness of pregnancy.** measured by the planning state of the last pregnancy, whether planned or not (i.e. unwanted or miss-timed pregnancies).

**Accessible distance to nearby health facility.** if a mother does not travel more than one hour on foot or by local means to reach healthcare facilities [39].

## Data analysis

The data was coded, cleaned, and entered by Epi-data software version 3.1 and exported to the SPSS version 23.0 statistical package for further analysis. Descriptive statistics such as frequency distributions, mean and standard deviation have been calculated to quantify the variables. The household's wealth status was analyzed using the Principal Components Analysis (PCA). Initially, 28 items composed of assets, livestock ownership, crop production in quintals, estimated average monthly income, farmland in hectares, and residential home with their infrastructure were considered. The asset or variables were excluded if they are owned by

more than 95% or less than 5% of the sample. The measurement of sampling adequacy by Kaiser-Meyer-Olkin (KMO≥0.6), the Bartlett Sphericity test (p-value < 0.05), and the anti-image correlations (> 0.4) for the suitability of the sampling of individual variables were checked if the PCA assumptions were met. Variables with communalities below 0.5 and complex structures (i.e. with correlations above 0.4 in more than one component) were removed at each step until the iterations met the criteria. Finally, three components that accounted for a total variance of 63.4% were extracted from the PCA and used to classify the household wealth status of study participants into quintiles [19].

A binary logistics regression model was used to identify factors associated with CPNC. Variables having a p-value lower than 0.25 in bivariate logistic regression were selected as candidates for multivariable logistic regression analysis. The adjusted odds ratio and its 95% CI were used to report the association of the explanatory variables with the outcome variable, and the statistical significance of each variable in the multivariable logistic regression model has been reported at a value less than 0.05.

### Ethical approval and consent to participate

Ethical clearance was obtained from the Institutional Review Board/IRB of Arba Minch University, College of Medicine and Health Science, School of Public Health. Interviewees were informed of the objective and procedure of the study. Written and oral consents were obtained from participants according to their level of education. To preserve the anonymity of the questionnaire, a unique identification number has been issued. Before data collection, participants' privacy and confidentiality were ensured. Respondents were also well informed that the information provided during the study would only be used for research purposes and would not be disclosed to anyone other than the research team. An official letter of permission was obtained from the Ezha District Health Department.

## Results

Of the 568 sampled women who gave birth in the past 12 months, 556 participated in the study, yielding a 97.8% response rate.

### Socio-demographic characteristics of respondents

Maternal age ranged from 18 to 42 years with a mean of 28.84(SD±5.306) years. Over two-fifths, (44.3%) of the study participants had no formal education. Nearly half (49.5%) of the participants were Orthodox Christians of religion, while Guraghe was the dominant ethnic group (93.2%) (Table 1).

### Obstetric characteristics of respondents

One half (50.4%) of study participants were multiparous. About nine out of ten mothers (89.2%) received at least one antenatal care (ANC) visit during their last pregnancy and 124 (22.3%) of them made four or more visits. Concerning the place of delivery 519(93.3%) of mothers gave their last birth at health facilities. Forty-one (7.4%) of the mothers gave birth to their last child through a cesarean section (Table 2).

### Maternal and neonatal health conditions during pregnancy and post-partum period

Over one-quarter (26.8%) of participants who reported at least one danger sign experienced a complication during their last pregnancy. Persistent vomiting 58 (38.9%) and a gush of fluid

**Table 1. Socio-demographic characteristics of respondents in Ezha district, Southern Ethiopia, February 10-March 10, 2019.**

| Respondent characteristics | Frequency | Percentage |
|---|---|---|
| **Maternal Age (n = 556)** | | |
| <20 | 19 | 3.4 |
| 20–34 | 437 | 79.1 |
| 35+ | 96 | 17.5 |
| **Maternal Educational status(n = 556)** | | |
| No formal education | 246 | 44.3 |
| Primary education | 167 | 30.0 |
| Secondary education | 125 | 22.5 |
| College and above | 18 | 3.2 |
| **Marital status (n = 556)** | | |
| Married | 530 | 95.3 |
| Single/divorced/widowed/unmarried | 26 | 4.7 |
| **Ethnicity (n = 556)** | | |
| Guraghe | 518 | 93.2 |
| Others | 38 | 6.8 |
| **Religion (n = 556)** | | |
| Orthodox | 275 | 49.5 |
| Muslim | 243 | 43.7 |
| Protestant | 38 | 6.8 |
| **Maternal occupation (n = 556)** | | |
| Housewife | 392 | 70.5 |
| Have private work | 113 | 20.3 |
| Daily laborer | 40 | 7.2 |
| Government employer | 11 | 2.0 |
| **Husband Educational status (n = 530)** | | |
| No formal education | 222 | 41.9 |
| Primary education | 194 | 36.6 |
| Secondary education | 93 | 17.5 |
| College and above | 21 | 4.0 |
| **Husband occupation (n = 530)** | | |
| Farmer | 380 | 71.7 |
| Merchant | 119 | 22.4 |
| Civil servant | 20 | 3.8 |
| Daily laborer | 11 | 2.1 |
| **Family size (n = 556)** | | |
| <5 | 283 | 50.9 |
| ≥5 | 273 | 49.1 |
| **Household wealth status(n = 556)** | | |
| Poorest | 119 | 21.4 |
| Poorer | 114 | 20.5 |
| Middle | 111 | 19.9 |
| Rich | 114 | 20.5 |
| Richest | 98 | 17.6 |

before the commencement of labor 31 (20.8%) were the primary complaints reported (Fig 1). Of the total number of respondents, 132 (23.7%) reported having had one or more maternal health issues during the post-partum period. The most common health problems reported by

**Table 2. Obstetric characteristics of respondents in Ezha district, Southern Ethiopia, February 10-March 10, 2019.**

| Respondents' Characteristics | Frequency | Percentage |
|---|---|---|
| **Parity (n = 556)** | | |
| Primiparous | 87 | 15.6 |
| Multiparous | 280 | 50.4 |
| Grand multiparous | 189 | 34.0 |
| **Previous history of neonatal death (n = 556)** | | |
| Yes | 45 | 8.1 |
| No | 511 | 91.9 |
| **Planning status of last pregnancy (n = 553)** | | |
| Intended | 438 | 79.2 |
| Unwanted | 42 | 7.6 |
| Mistimed | 73 | 13.2 |
| **ANC visits (n = 556)** | | |
| No | 60 | 10.8 |
| One visit | 165 | 29.7 |
| Two visits | 94 | 16.9 |
| Three visits | 113 | 20.3 |
| Four and more visits | 124 | 22.3 |
| **Last Birth outcome (n = 556)** | | |
| Live birth | 540 | 97.1 |
| Stillbirth | 16 | 2.9 |
| **Place of delivery (n = 556)** | | |
| Health center | 424 | 76.2 |
| Hospital | 112 | 20.1 |
| Health post | 20 | 3.7 |
| **Mode of delivery(n = 556)** | | |
| Cesarean delivery | 41 | 7.4 |
| Instrumental delivery | 53 | 9.5 |
| Spontaneous vaginal delivery (SVD) | 462 | 83.1 |

respondents were breast problems (31.8%), abnormal vaginal bleeding (34.4%), high-grade fever (26.2%), severe abdominal pain (7.7%), and swelling of body parts (2.5%). On the other hand, a quarter (25.0%) reported that they had experienced at least one neonatal illness during the post-partum period. Failure to breastfeed (53.1%) and difficulty of breathing (30.4%) were the primary complaints of newborns, followed by umbilical area infections (11.7%), vomiting (9.4%), and high-grade fever (8.6%).

## Level of knowledge of respondents on postnatal care service

A composite knowledge indicator was constructed, according to which 136(24.5%), 288 (51.8%), and 132 (23.7%) of women had a poor, moderate, and good knowledge of PNC services. For information exposure, 355 (63.8%) of respondents heard about the PNC service. Nearly three-quarters (74.9%) and 156 (43.9%) obtained information from health extension workers and health care providers at health facilities, respectively. Additionally, 121 (34.1%) and 31 (8.7%) of respondents obtained information from friends and radio, respectively. Close to half (49.2%) of mothers were aware of at least one benefit of receiving postnatal care. Approximately one-quarter (23.3%) of respondents were aware of the duration of PNC service

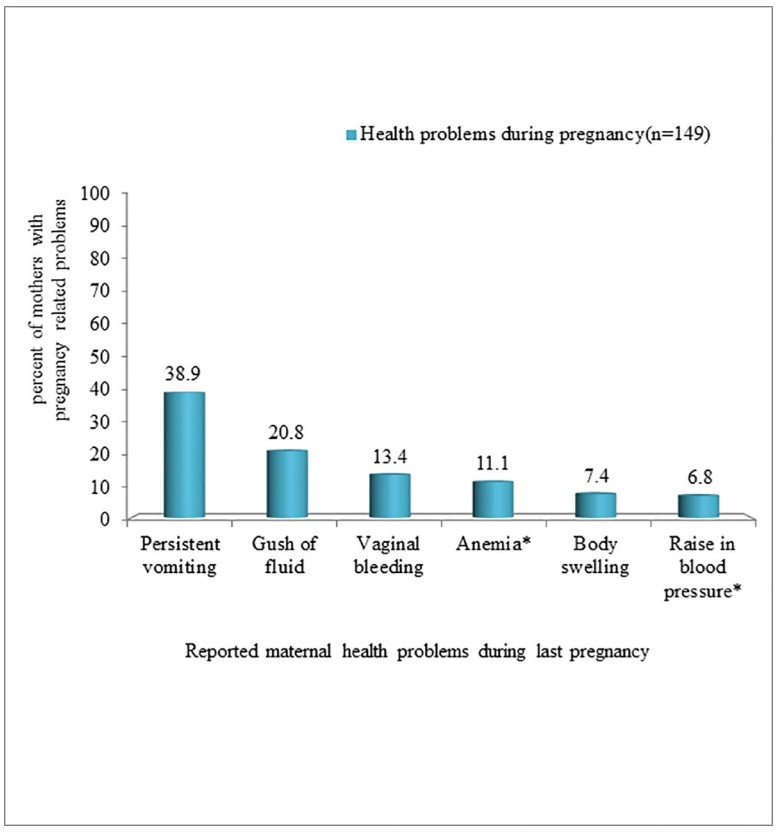

**Fig 1. Danger signs reported by study participants during their last pregnancy in Ezha district, Southern Ethiopia, February10-March 10 2019.**

delivery (i.e. after delivery and up to six weeks), while only 102 (18.4%) were aware of the recommended frequency of post-natal visits. As well, 54.9% and 57.4% of mothers can mention at least one danger sign in mothers and newborns during postpartum periods, respectively.

## Health system-related characteristics of respondents

The average time to walk to a nearby healthcare facility was 52.4 minutes and 361 (64.9%) of respondents accessed a healthcare facility within an hour. For autonomy over the use of maternal health services, the majority (87.4%) of respondents were autonomous (Table 3).

## Uptake of complete postnatal care (CPNC) services

From a total of 556 respondents, 109 [23.9%; 95%CI: (19.9, 27.5)] received all four postnatal visits by skilled health care providers. Regarding the timing of PNC visits, 392(70.5%) mothers received their postnatal contact within the first 24 hours of delivery. Also, 296(53.2%), 248 (44.1%) and 152(27.3%) respondents received a postnatal visit on 3–6 days, 7–14 days, and at the sixth week respectively. Regarding the service delivery platform, nearly three-quarters quarter (74.9%) respondents got the service by health extension workers through the home-to-home visit and/or at health post level while the remaining (25.1%) got the service at the health center and hospital level. Long-distance travel to health facilities was the main reason for not complying with recommended PNC visit schedules (Fig 2).

**Table 3. Health system-related characteristics of study participants in Ezha district, Southern Ethiopia, February 10-March 10, 2019.**

| Health system-related characteristics | Frequency | Percent |
|---|---|---|
| **Autonomy to maternal health services(n = 556)** | | |
| Self-decision | 263 | 47.3 |
| Joint-decision | 223 | 40.1 |
| Husband only/Third party | 70 | 12.6 |
| **Got help/support to maternity service(n = 556)** | | |
| Husband | 453 | 81.5 |
| Relatives | 94 | 16.9 |
| Others | 9 | 1.6 |
| **Being a model household(n = 556)** | | |
| Yes | 269 | 48.4 |
| No | 277 | 51.6 |

## Contents of care given during PNC service delivery

In terms of the content of the PNC services, 390 (70.1%) of respondents received health education on exclusive breastfeeding. Relatively few respondents received iron folate supplements (20.1%) and a breast examination (15.0%) (Table 4). There was considerable variation in the receipt of recommended content of care between mothers with complete PNC and their counterparts. Of the 109 respondents who obtained a complete PNC, the majority (91.7%) were counseled about exclusive breastfeeding and 102 (93.5%) were examined for abnormal vaginal bleeding. On the other hand, most of these participation rates were low among respondents who had a sporadic PNC (Fig 3).

## Determinants of complete postnatal care utilization

A multivariable logistic regression analysis was carried out and five variables, namely maternal education, antenatal care frequency, cesarean delivery, mothers' knowledge of PNC, and being a model household, were significantly associated with the use of CPNC.

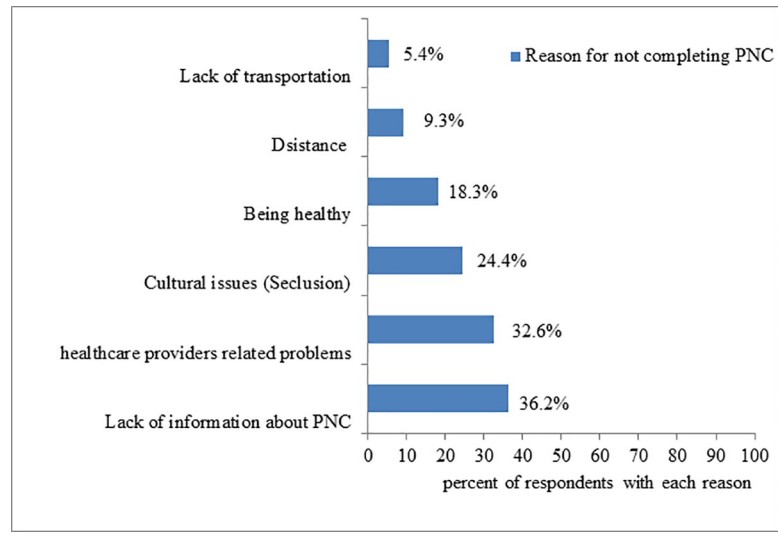

**Fig 2. Reasons for not completing recommended PNC visits among women of Ezha district, Southern Ethiopia, February 10-March 10, 2019.**

**Table 4. WHO recommended contents of care provided to the study participants during their postnatal visit Ezha district, southern Ethiopia, February 10-March 10, 2019.**

| Contents of care(n = 556) | Frequency | Percentage |
|---|---|---|
| Health education on exclusive breastfeeding (EBF) | 390 | 70.1 |
| Examination of abnormal vaginal bleeding (AVB) | 386 | 69.4 |
| Counseling on danger signs | 349 | 62.8 |
| Counseling on personal hygiene | 228 | 41.0 |
| Health education on contraception | 312 | 56.1 |
| Health education on HIV/AIDS transmission | 206 | 37.1 |
| Iron folate supplementation(IFAS) | 112 | 20.1 |
| Cord care is given to the newborn by using chlorhexidine | 294 | 52.8 |
| Provision of eye care with Tetracycline(TTC) | 143 | 25.8 |
| Breast examination | 83 | 14.9 |
| Measuring body temperature of the newborn | 254 | 45.7 |
| Getting referral service | 92 | 16.5 |
| Measuring body weight of the newborn | 242 | 43.5 |

In comparison to mothers lacking formal education, the odds of receiving at least three PNC visits are 4.3 times higher for those pursuing secondary education and above (AOR: 4.30, 95%CI: 2.15–8.05]. Similarly, the odds of receiving CPNC are 3.7 times higher for mothers who have received four or more ANC visits than for those who have received at most one visit (AOR: 3.75, 95% CI: 1.78–7.92). Being a model household was significantly associated with receiving full PNC visits. Compared to their counterparts, the chances of receiving CPNC among mothers from the model household were 3.6 times greater (AOR: 3.61, 95% CI: 1.97–6.64). In contrast to those with poor knowledge, women with good knowledge of postnatal care also had a higher chance of complying with PNC visit schedules (AOR: 4.31, 95% CI: 2.34–9.04) (Table 5).

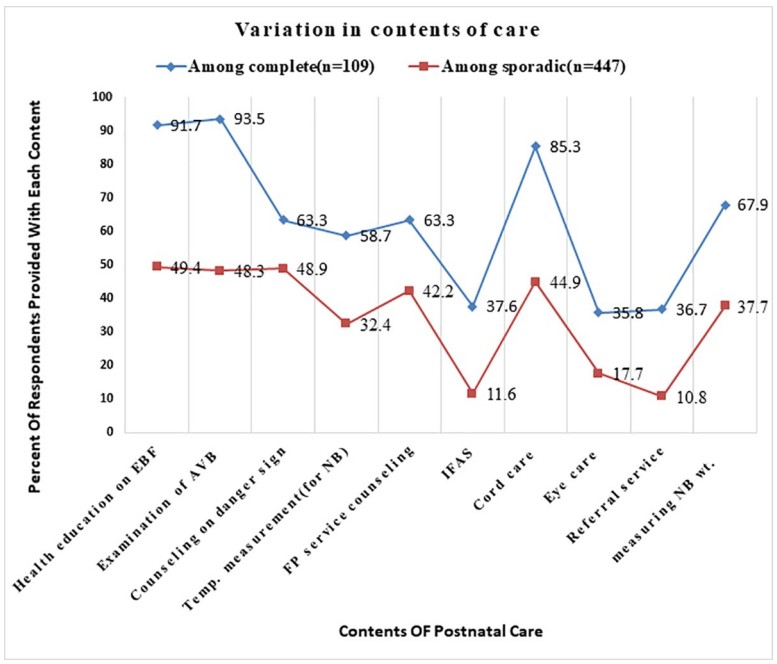

**Fig 3. Variation in the receipt of contents of care among study participants with complete versus sporadic PNC, in Ezha district, Southern Ethiopia, February 10-March 10, 2019.**

**Table 5. Factors associated with the level of CPNC service utilization among study participants in Ezha district, Southern Ethiopia, February 10-March 10, 2019.**

| Variable categories | CPNC status | | COR(95%CI) | AOR(95%CI) | p-value |
|---|---|---|---|---|---|
| | Yes (%) | No (%) | | | |
| **Maternal Educational status (n = 556)** | | | | | |
| No formal education | 16(6.5) | 230(93.5) | 1 | 1 | 1 |
| Primary education | 43(25.7) | 124(74.3) | 4.98(2.69,9.21) [a] | 2.46(1.22,4.92) [b] | 0.097 |
| Secondary and above | 50(34.9) | 93(65.1) | 7.73(4.19,14.26)[a] | 4.30(2.15,8.05) [b] | <0.001 |
| **Maternal age(n = 556)** | | | | | |
| 35+ | 5(5.2) | 91(94.8) | 1 | 1 | 1 |
| 20–34 | 98(22.3) | 342(77.7) | 5.21(2.06,13.19)[a] | 2.25(0.78,6.50) | 0.061 |
| <20 | 6(30.0) | 14(70.0) | 7.80(2.09,21.01)[a] | 2.09(0.35,12.29) | 0.915 |
| **Family size(n = 556)** | | | | | |
| ≥5 | 40(14.6) | 233(85.7) | 1 | 1 | 1 |
| <5 | 69(24.4) | 214(75.6) | 1.88(0.94, 2.89) | 0.67(0.38,1.18) | 0.295 |
| **Planning status of last pregnancy (n = 553)** | | | | | |
| Planned | 102(23.3) | 336(76.7) | 4.68(2.11, 9.38) [a] | 2.34(0.92,5.18) | 0.817 |
| Unplanned [c] | 7(6.1) | 108(93.9) | 1 | 1 | 1 |
| **Number of ANC visits (n = 556)** | | | | | |
| ≤ one visit | 15(6.7) | 210(93.3) | 1 | 1 | 1 |
| Two visits | 17(18.1) | 77(81.9) | 3.09(1.47,6.49) [a] | 2.18(0.92,5.14) | 0.147 |
| Three visits | 31(27.4) | 82(72.6) | 5.29(2.71,10.31) [a] | 2.49(1.15,5.39) [b] | 0.020 |
| Four or more | 46(37.1) | 78(62.9) | 8.25(4.36,15.63) [a] | 3.75(1.78,7.92) [b] | 0.001 |
| **Parity(N = 556)** | | | | | |
| Grand multiparous | 20(10.6) | 169(89.4) | 1 | 1 | 1 |
| Multiparous | 61(21.8) | 219(78.2) | 2.35(0.97,4.05) | 1.26(0.51,3.07) | 0.062 |
| Primiparous | 28(32.2) | 59(67.8) | 4.01(2.1,7.65) [a] | 1.24(0.58,2.64) | 0.086 |
| **Mode of delivery (n = 556)** | | | | | |
| SVD | 71(15.4) | 391(84.6) | 1 | 1 | 1 |
| Instrumental | 20(37.7) | 33(62.3) | 3.34(1.81,6.14) [a] | 1.57(0.76,3.26) | 0.057 |
| C/S | 18(43.9) | 23(56.1) | 4.31(2.213,8.39) [a] | 3.96(1.5,7.94) [b] | 0.005 |
| **Knowledge of PNC (n = 556)** | | | | | |
| Poor | 12(8.8) | 124(91.2) | 1 | 1 | 1 |
| Moderate | 52(18.1) | 236(81.9) | 2.27(1.17,4.42) [a] | 1.87(1.00,4.70) | 0.71 |
| Good | 45(34.1) | 87(65.9) | 5.34(2.67,10.67)[a] | 4.31(2.34,9.04) [b] | 0.007 |
| **Time To reach nearby HF** | | | | | |
| Far (>1hr) | 27(13.8) | 168(86.2) | 1 | 1 | 1 |
| Close (< = 1hr) | 82(22.7) | 279(77.3) | 1.83(0.78, 2.94) | 1.17(0.63,2.19) | 0.095 |
| **Being a model household (MHH) (n = 556)** | | | | | |
| Yes | 91(33.8) | 178(66.2) | 7.64(4.45,13.1) [a] | 3.61(1.97,6.64)[b] | <0.001 |
| No | 18(6.3) | 269(93.7) | 1 | 1 | 1 |

1: reference category; COR = Crude Odds Ratio, AOR = Adjusted odds ratio, MHH = Model Household

[a] significant in a bivariate analysis at a p-value of less than0.25,

[b] statistically significant association at a p-value of less than 0.05.

## Discussion

This study found that 23.9% of mothers in the district obtained a complete post-natal care service (CPNC). Maternal education, antenatal care frequency, cesarean delivery, mothers' knowledge of PNC, and being model households have been significantly associated with the use of the CPNC services.

The use of CPNC service was consistent with a report from a study conducted in Myanmar [9] and lower than those conducted in India, rural Ghana (Builsa and Mamprusi districts), and northern Ethiopia [20, 22, 23]. The low use of CPNC in this study may be due to a disparity in the study area where all women in this study were rural residents who did not have better access to healthcare, information, and education. On the other hand, the result was higher than the analysis of the survey carried out in three rural districts of Tanzania for the full PNC [22]. The discrepancy could be attributed to the time gap, as there would be improved access to health care and service awareness over time.

Furthermore, the level of use of the CPNC is significantly lower among study participants relative to the proportions of ANC, general PNC of at least one visit, and skilled delivery services. As such, local health departments need to plan in accessing these rural women with CPNC services through strengthening outreach programs and home to home visits during the postpartum period. Women's educational attainment was significantly associated with the use of the CPNC. Higher probabilities of CPNC use were found among women who attended secondary education and above compared to women without formal education. Evidence from studies in Myanmar, India, Tanzania, and northern Ethiopia supports this finding [9, 20, 22, 23]. This could be explained by the idea that education is one of the key factors that improve maternal autonomy towards improved health-seeking behavior of maternal and child health services [38, 40]. Furthermore, mothers with a higher level of education may be more aware of the importance of obtaining PNC service due to the possibility of being exposed to various sources of information with better information processing skill, this may lead mothers to use the CPNC as compared to their counterparts. This could be well supported by the current study that 85.3% of mothers who received CPNC had formal education and 45.9% attended secondary school and beyond.

The frequency of antenatal care is one of the factors positively associated with the use of the CPNC. Women who received four or more antenatal care visits were more likely to receive CPNC. This finding complemented a previous study in Ghana [21]. The possible rationale for this finding may be that adequate antenatal care provides women with access to health promotion on danger signs and postpartum complications, which leads to subsequent use of services [11, 41]. Furthermore, women who attend the ANC more frequently are more likely to receive advice on the importance, availability, timing, and adherence to the recommended PNC and may end up with the use of the CPNC. In the current study, 70.6% of CPNC uptake was encountered by those mothers with three and more visits. We, therefore, suggest that the beginning of the ANC is essential to the adoption of the CPNC and that the District Health Taskforce should focus on the timely initiation and completion of the ANC service.

Women's knowledge of postnatal care was one factor that increased the likelihood of CPNC utilization. This finding has been corroborated in studies conducted elsewhere [9, 20, 34]. This is plausible because the more adequate postnatal care knowledge a mother had (such as its benefits, care content, the timing of the visit, the consequence of not receiving the service, postpartum danger signs for maternal and newborn); the more she might comply with the recommended PNC visits. Moreover, being well informed may be an important factor in motivating women and their families to take up the service. These justifications could be supported by the present study in which 60% of respondents who use CPNC were knowledgeable. Therefore, a concerted effort is needed by the concerned bodies to improve women's knowledge of the PNC so that women and newborns can benefit from the use of the CPNC service.

The likelihood of using CPNC was higher for women from model households compared to their counterparts in this study. This can be supported by a study from Ethiopia [42]. This may be because Health Extension Workers (HEWs) spend more time on capacity building for model HHs through intensive training, support, and follow-up with practical demonstration

and family education on maternal and child health services for those identified to be role models [43, 44]. Furthermore, those women from model households might have an opportunity to involve in larger community meetings where all residents in a community will participate regularly. These larger public conferences provide a platform to discuss prioritized bottlenecks and strategies regarding basic maternal health services and might make them complied with recommended PNC services. Results from previous studies have shown that model families use maternal health services effectively [42, 44, 45]. This may be well supported by the result of this study showing that 83.5% of mothers who received a CPNC were from model households. Hence, district health workers must plan for the creation of model households to enhance the adoption of CPNC and other MNCH services.

In addition to the primary objective, an attempt was made to evaluate the content of care during the PNC visits and most of the figures were higher compared to studies conducted in Myanmar and China [9, 46]. According to a study in Myanmar, 48.1%, 51.6%, and 49.3% of mothers received information on exclusive breastfeeding, postnatal danger signs, and methods of contraception, respectively [9]. These figures for this study were 70.1%, 62.6%, and 56.1%, respectively. Similarly, in China where 37%, 32%, and 18% of respondents got an education on EBF, cord care, and counseling on danger signs respectively [46]. These differences in the content of care could be attributed to variation in maternal knowledge about PNC, use of maternal health care (ANC, delivery, and early PNC), in which the majority of these services were higher in the current study area. Conversely, the Myanmar study found that almost all mothers (98.8%) were receiving postnatal iron supplementation which is too lower in the current study area (20.1%). We, therefore, recommend that post-natal iron supplementation be strengthened within the current study area.

There were both strengths and shortcomings of this study. There was insufficient evidence about CPNC and its determinants based on the latest WHO guideline (2013) [7]. So it could be used locally and at the policy level as an input. Since this study is cross-sectional, there was no relationship between cause and effect reported. Furthermore, despite consideration of more recent births, it can be difficult to recall the exact timing of visits and the content of care provided by qualified health care providers during the postnatal period due to recall bias. Lastly, the study was based on self-reporting, which could have resulted in a social desirability bias.

## Conclusion

The study concluded that the level of use of CPNC was lower than in other studies. The mother's education level, the frequency of ANC visits, the mode of delivery, postnatal care knowledge, and being a model household were factors that are significantly associated with the use of CPNC in the study area. Thus, communication activities on behavioral change to improve maternal knowledge about PNC, Improving access to adequate ANC, and working on the creation of model households should be critical measures to improve the use of CPNC in the study area. The District Education Office needs to do everything possible to encourage girls and women to pursue higher education.

## Supporting information

**S1 Dataset. The raw data supporting the findings of this article.**
(SAV)

**S1 Questionnaire. Data collection tool for the study.**
(DOCX)

## Acknowledgments

We are indebted to Arba Minch University College of Medicine and Health Science, Department of Public Health for giving Ethical clearance to undertake the study. Our appreciation also goes to the managers and healthcare providers who worked in the Ezha district Health Office for their assistance and cooperation during the study. Finally, for their efforts, we want to thank our supervisors, data collectors, and study participants.

## Author Contributions

**Conceptualization:** Aklilu Habte.

**Data curation:** Misgun Shewangizaw, Samuel Dessu, Mustefa Glagn.

**Formal analysis:** Aklilu Habte.

**Investigation:** Feleke Gebiremeskel.

**Methodology:** Aklilu Habte, Feleke Gebiremeskel, Samuel Dessu, Mustefa Glagn.

**Resources:** Misgun Shewangizaw.

**Software:** Aklilu Habte, Samuel Dessu.

**Supervision:** Aklilu Habte, Feleke Gebiremeskel, Misgun Shewangizaw, Samuel Dessu, Mustefa Glagn.

**Writing – original draft:** Aklilu Habte.

**Writing – review & editing:** Aklilu Habte.

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
