## [Decision Letter · Decision Letter 0]

29 Jul 2020

PONE-D-20-05045

Complete postnatal care utilizations and its associated factors among women who gave birth in the last 12 months in Ezha district, Southern Ethiopia , 2019: Community-based cross-sectional study: Compliance to the current WHO recommendation

PLOS ONE

Dear Dr. Habte,

Thank you for submitting your manuscript to PLOS ONE. After careful consideration, we feel that it has merit but does not fully meet PLOS ONE’s publication criteria as it currently stands. Therefore, we invite you to submit a revised version of the manuscript that addresses the points raised during the review process.

We look forward to receiving your revised manuscript.

Kind regards,

Onikepe Oluwadamilola Owolabi

Academic Editor

PLOS ONE

Journal Requirements:

2. Please specify in your ethics statement whether participant consent was written or verbal. If verbal, please also specify: 1) whether the ethics committee approved the verbal consent procedure, 2) why written consent could not be obtained, and 3) how verbal consent was recorded.

Reviewers' comments:

Reviewer's Responses to Questions

**Comments to the Author**

1. Is the manuscript technically sound, and do the data support the conclusions?

Reviewer #1: No

Reviewer #2: Yes

2. Has the statistical analysis been performed appropriately and rigorously? 

Reviewer #1: No

Reviewer #2: No

3. Have the authors made all data underlying the findings in their manuscript fully available?

Reviewer #1: No

Reviewer #2: Yes

4. Is the manuscript presented in an intelligible fashion and written in standard English?

Reviewer #1: Yes

Reviewer #2: No

5. Review Comments to the Author

Reviewer #1: The authors measured PNC based on the WHO definition which takes timing into consideration. How did the authors treat births at home? Were they dropped from the analysis? This needs to be mentioned in this section. It is also important to mention the type of personnel.

Not convinced with the significance of the study.

Reviewer #2: Title of the article Complete postnatal care utilizations and its associated factors among women who gave birth in the last 12 months in Ezha district, Southern Ethiopia, 2019: Community- based cross-sectional study: Compliance to the current WHO recommendation

The subject is relevant and brings current and important information to the context of maternal and neonatal health in Africa. The findings are substantial and may contribute to health policies implementation in low- and middle-income settings. However, the text needs several improvements in order to grant proper discussion and analysis of included issues.

Background

The English of the manuscript needs significant proofreading.

The importance of complete postnatal care utilization is not well emphasised. As you mentioned on page 3 and 3rd paragraph, “within the first week of life increases…”. What’s the rationale to compare the complete PNC? Where all other PNC utilized are categorized as sporadic/non-case?

Methodology

On the data analysis section “continuum of care was mentioned”?

Result

Page 13 on the prevalence of complete postanal care utilization section- those mothers who never sought PNC were overlooked and not presented. I was wondering if mothers with home delivery were in your exclusion criteria. Also, considering the national facility delivery proportion, the number of mothers who sought 1st PNC service are too high according to your findings. Could you comment that on the manuscript?

On the outcome variable – Sporadic vs complete postnatal care- mothers who don’t get postnatal care are overlooked in this research.

Was PNC considered after discharge for CS? Or while women were admitted? On the wealth status categories consider poorest to richest instead of lowest to highest

Include more detailed limitations of the study. Causal relationship is impossible in this study design, not difficult. What type of recall bias is likely? What women were missing from the study, e.g. Is it possible that the CHEW birth registers missed women and these women would be the most vulnerable and least likely to get PNC? What about women with stillbirths or early neonatal deaths? were they included? What difference would it make

Discussion

The discussion is full of results already displayed and lacks depth in addressing the actual findings. In addition to quoting and comparing previous studies with current results, reflections on present findings should be conducted, e.g. addressing gender issues. The ideas are disclosed, however could be further elaborated in order to improve the manuscript's strength.

6. PLOS authors have the option to publish the peer review history of their article (what does this mean?). If published, this will include your full peer review and any attached files.

Reviewer #1: No

Reviewer #2: No

---

## [Author Response · Author response to Decision Letter 0]

21 Aug 2020

AUTHORS’ RESPONSE TO EDITOR AND REVIEWERS

I. AUTHORS’ RESPONSE TO EDITOR

Dear: Onikepe Oluwadamilola Owolabi

We thank you and the reviewers for a thorough reading and constructive criticism of our manuscript and for the opportunity to revise and resubmit. We are pleased to submit the revised research article “Uptake of Complete postnatal care services and its determinants among women in Ezha district, Guraghe zone, Southern Ethiopia, 2019: Community-based cross-sectional study based on the current WHO recommendation” for your consideration in the special collection of PLOS ONE. On the following pages, you will find our response to the editor’s comments. On behalf of my co-authors, I thank you for your consideration of this resubmission. We appreciate your time and look forward to your response.

Sincerely, 

Aklilu H.(MPH)(corresponding author)

aklilihabte57@gmail.com

General comments

Comment from The Review Process: Thank you for submitting your manuscript to PLOS ONE. After careful consideration, we feel that it has merit but does not fully meet PLOS ONE’s publication criteria as it currently stands. Therefore, we invite you to submit a revised version of the manuscript that addresses the points raised during the review process.

RESPONSE: Thank you for coordinating the review process. We have revised the manuscript by exploiting all our maximum efforts and tried to address Reviewer’s comments in advance.

 Comment1: Please ensure that your manuscript meets PLOS ONE's style requirements, including those for file naming. The PLOS ONE style templates can be found at https://journals.plos.org/plosone/s/file?id=wjVg/PLOSOne_formatting_sample_main_body.pdf andhttps://journals.plos.org/plosone/s/file?id=ba62/PLOSOne_formatting_sample_title_authors_affiliations.pdf

Response: After accessing those file templates we have checked and attest that all formatting style of the current manuscript and now we reformatted the manuscript accordingly and all these have been highlighted in the “revised manuscript with track changes’’

Comment2- Please try to be specific in your ethics statement whether participant consent was written or verbal. If verbal, please also specify: 1) whether the ethics committee approved the verbal consent procedure, 2) why written consent could not be obtained, and 3) how verbal consent was recorded.

Response: Thank you for your comment. Of course, the initial manuscript lacks detail explanation of the ethics statement. Now we have revised it and included ethical procedures that were not mentioned in the original manuscript in 3rd line of “Ethical approval and consent to participate” and this has been highlighted and underlined in the “revised manuscript with track changes” file. The ethics committee approved the verbal consent procedure.

Comment3- Please include additional information regarding the survey or questionnaire used in the study and ensure that you have provided sufficient details that others could replicate the analyses. For instance, if you developed a questionnaire as part of this study and it is not under a copyright more restrictive than CC-BY, please include a copy, in both the original language and English, as Supporting Information.

Response: We really appreciate for your critical comment. Actually we have mentioned about the how the tool was developed and employed for the study in the 2nd line of “Data collection tools, methods and procedure”. It was developed by reviewing relevant literatures. Now we have incorporated both English and Amharic (original language) version of the questionnaire as supportive information, “S1-Questionnaire”

Comment4- Please amends your list of authors on the manuscript to ensure that each author is linked to an affiliation. Authors’ affiliations should reflect the institution where the work was done (if authors moved subsequently, you can also list the new affiliation stating “current affiliation:….” as necessary).

Response: Thank you for the comment. Now we have re-arranged affiliation of the authors with their current specific department and units where they are working. The amendments were based on the specific working unit of each author with their level of responsibility in the study. This has been highlighted in Pge1, line1of “revised manuscript with track changes” file.

END________________________________________

II. AUTHOR’S RESPONSE TO REVIEWERS’ COMMENTS

Dear: Reviewers

We thank you for a thorough reading and constructive criticism of our manuscript and for the opportunity to revise and resubmit. We are pleased to submit the improved research article “Uptake of Complete postnatal care services and its determinants among women in Ezha district, Guraghe zone, Southern Ethiopia, 2019: Community-based cross-sectional study based on the current WHO recommendation” for your consideration in the special collection of PLOS ONE. On the following pages, you will find our response to the reviewers’ comments. On behalf of my co-authors, I thank you for your consideration of this resubmission. We appreciate your time and look forward to your response independently as for reviewer#1 and reviewer#2 respectively.

Sincerely, 

Aklilu H. (MPH)(corresponding author)

aklilihabte57@gmail.com

AUTHOR’S RESPONSE TO REVIEWER#1

General comments

Comment1-These issues are related to technical soundness of the manuscript and the conclusions part

Response to reviewer1: We really appreciate for your view. After your comments and suggestions we gave due emphasis in order to make the manuscript technically sound from its write up to detail methodology and analysis part. To assure representativeness, the sample size was determined for both first and second objective and we used the maximum sample size for the study. Regarding to the conclusion part, it’s based on our findings from the study without ignoring our initial objectives. But we kindly informing you that there were some statements which were omitted from the conclusion part of revised manuscript in order to avoid redundancy and those omitted narrations were already mentioned in the descriptive part of the result. This has been highlighted and underlined in the conclusion part of “revised manuscript with track changes” file.

Comment2- this issue is regarding to appropriateness of the statistical analysis been performed appropriately and rigorously?

Reviewer #1: NO

Response: We really appreciate your helpful examination of these errors. We are convinced with your concern, and the level of complete postnatal care is assessed in terms of those women who get at least one postnatal visit, in order to avoid overlooking of those mothers with no PNC. As reviewer 2 has also a similar concern, we have further clarified this in the result part and highlighted in the 2nd to 3rd line of “Level of complete post natal care utilization’’ of revised manuscript with track changes’’ file. Let us explain why this error has been committed. Initially we perceived that excluding those mothers without PNC from the analysis mayn’t cause significant difference in the level of CPNC because of their relatively small count. But now we are in agreement with your concern and corrected it accordingly throughout the current manuscript.

Comment3-Have the authors made all data underlying the findings in their manuscript fully available? (Reviewer1= NO)

Response: We appreciate the comment and we have uploaded our supplementary information including the full data set, English and Amharic (local language) version of the questionnaire and result of bivariate analysis in the current submission.

Comment4- Regarding presentation of the manuscript in an intelligible fashion and written in Standard English (Reviewer1=Yes).

Response: thank you for your constructive and positive view regarding the manuscript.

Comment5- The authors measured PNC based on the WHO definition which takes timing into consideration. How did the authors treat births at home? Were they dropped from the analysis? This needs to be mentioned in this section. It is also important to mention the type of personnel.

Response: We really appreciate for your view. Regarding home at births, they were not dropped from the analysis. Initially during conceiving of the study we were planned to put home births in exclusion criteria, but we let them to be part of the study for two reasons:

1. Currently, every Kebele of the district have adequate number of health extension workers to its population and once home delivery has been occured the health extension worker/s immediately accessed that woman within 24 hours of duration and she might have a probability of receiving the PNC services. This was a little bit supported by the study in which 2 mothers with home delivery got complete post natal care service.

2. The institutional review board also raised the issue of excluding those mothers with home birth from the study as unethical issue.

END________________________________________

AUTHOR’S RESPONSE TO REVIEWER#2

General comments

Comment1-These issues are related to technical soundness of the manuscript and the conclusions part (Reviewer2=YES)

Response: We really appreciate for your positive view. 

Comment2- this issue is regarding to appropriateness of the statistical analysis been performed appropriately and rigorously?

Reviewer #2: No

Response: We really appreciate your helpful examination of these errors. Let us explain why this error has been committed. Initially we thought that excluding those mothers without PNC from the analysis mightn’t cause significant difference in the level of CPNC because of their relatively small count. We are convinced with your concern, that the level of complete postnatal care is assessed in terms of those women who get at least one postnatal visit, in order to avoid overlooking of those mothers with no PNC. As reviewer 1 have also a similar concern, we have further clarified this in the result part and highlighted in the “Level of complete post natal care utilization’’ of revised manuscript with track changes’’ file. But now we are in agreement with your concern and corrected it accordingly in the current manuscript.

Comment3-Have the authors made all data underlying the findings in their manuscript fully available? (Reviewer2=Yes)

Response: thank you for the positive comment. 

Comment4- Regarding presentation of the manuscript in an intelligible fashion and written in Standard English.

Response: thank you for your constructive comment. First we are in agreement with your view and we gave due emphasis and endeavor to solve the issue throughout the manuscript. 

Comment5-Title of the article complete postnatal care utilizations and its associated factors among women who gave birth in the last 12 months in Ezha district, Southern Ethiopia, 2019: Community- based cross-sectional study: Compliance to the current WHO recommendation. The subject is relevant and brings current and important information to the context of maternal and neonatal health in Africa. The findings are substantial and may contribute to health policies implementation in low- and middle-income settings. However, the text needs several improvements in order to grant proper discussion and analysis of included issues.

Response: We would like to thank the reviewer for your kind words and supportive ideas about the current study as it is a contemporary issue. We had given a due emphasis to make improvements in each and every section of the manuscript accordingly.

Comment6-The English of the manuscript needs significant proofreading.

Response: We have revised as per your recommendations and we tried to manage those issues in-depth. We have tried to show most of these amendments in the “revised manuscript with track changes”

Comment7-The importance of complete postnatal care utilization is not well emphasized. As you mentioned on page 3 and 3rd paragraph, “within the first week of life increases…”. What’s the rationale to compare the complete PNC? Where all other PNC utilized are categorized as sporadic/non-case?

Response7.1: We really appreciate your helpful examination. There were shortage of literatures that directly written on complete postnatal care. But we are trying to address the importance of CPNC in terms adequate PNC visit schedules and were mentioned in Page4Paragraph1, 1st -5th lines and last paragraph of ‘Introduction’ part of revised manuscript.

Response7.2: “within the first week of life increases…”. Now this is corrected as “within the first six weeks” in page, 3 line23 of revised manuscript.

Response7.3: Thank you for your concern. When we come to our reason for comparison, every visit schedule has its own importance in provision of contents of care. For example last visit schedule at sixth week is important for the woman and newborn in initiating for vaccination, family planning and health educations regarding to essential nutrition actions. Now we are comparing provision of contents of care between mothers with complete and sporadic PNC and we appreciate huge differences in between. So, this might be clearly showed that importance of receiving CPNC to get adequate contents of care, this in turn support the importance of the current study. 

Comment8- “On the data analysis section “continuum of care was mentioned”?

Response8: sorry for the inconvenience that’s wrongly written and now it is corrected as “CPNC” in 24th line of “data analysis” in “revised manuscript with track changes”.

Comment9- Page 13 on the prevalence of complete postnatal care utilization section- those mothers who never sought PNC were overlooked and presented. I was wondering if mothers with home delivery were in your exclusion criteria. Also, considering the national facility delivery proportion, the number of mothers who sought 1st PNC service is too high according to your findings. Could you comment that on the manuscript?

Comment9.1: Page 13 on the prevalence of complete postnatal care utilization section- those mothers who never sought PNC were overlooked and presented.

Response9.1: we are greatly appreciating your comment. We had tried to address this issue in response 2 of your comment2. Initially, we thought that excluding those mothers without PNC from the analysis mightn’t cause significant difference in the level of CPNC because of their relatively small count. But now, we are in agreement with your concern, that the level of complete postnatal care is assessed in terms of those women who get at least one postnatal visit, in order to avoid overlooking of those mothers with no PNC. As reviewer 1 have also a similar concern, we have further clarified this in the result part and highlighted in the 1st to 3rd line of “Level of complete post natal care utilization’’ of revised manuscript with track changes’’ file. 

Comment9.2-I was wondering if mothers with home delivery were in your exclusion criteria.

Response9.2: We really appreciate for your view. Regarding home deliveries, they were part of the study and not dropped from the analysis. Initially during conceiving of the study we were planned to put home births in exclusion criteria, but we let them to be part of the study in two reasons:

1. Currently, every Kebele of the district have adequate number of health extension workers to its population and once home delivery has been happened the health extension worker/s immediately accessed that woman within 24 hours of duration and she might have a probability of receiving the PNC services. This was a little bit supported by the study in which 2 mothers with home delivery got complete post natal care service.

2. The institutional review board also raised the issue of excluding those mothers with home birth from the study as unethical issue.

 Comment9.3-Also, considering the national facility delivery proportion, the number of mothers who sought 1st PNC service is too high according to your findings. Could you comment that on the manuscript?

Response9.3: in comparison with report of EDHS2016 and mini-EDHS2019 report, all maternal health services in the study area were with higher coverage. We also noticed that high PNC1 in the current study, this might be due to high proportion skilled delivery service in the study area. 

Comment10- On the outcome variable – Sporadic vs complete postnatal care- mothers who don’t get postnatal care are overlooked in this research.

Response10: we are greatly appreciating your comment. We had tried to address this issue in response 9.1 of your comment9. Initially, we thought that excluding those mothers without PNC from the analysis mightn’t cause significant difference in the level of CPNC because of their relatively small count. But now, we are in agreement with your concern, that the level of complete postnatal care is assessed by taking of those mothers who get at least one postnatal visit, in order to avoid overlooking of those mothers with no PNC. As reviewer 1 have also a similar concern, we have further clarified this in the result part and highlighted in the 1st to 3rd line of “Level of complete post natal care utilization’’ of revised manuscript with track changes’’ file.

Comment11- Was PNC considered after discharge for CS? Or while women were admitted?

Response11: in this study all visits other than recommended PNC schedules (like maternal and newborn illness including C/S) were not considered as PNC service. 

Comment12- the wealth status categories consider poorest to richest instead of lowest to highest

Response12: this was corrected accordingly and highlighted in the result part page14, table 1 of” revised manuscript with track changes” file.

Comment13- Include more detailed limitations of the study. Causal relationship is impossible in this study design, not difficult. What type of recall bias is likely? What women were missing from the study, e.g. Is it possible that the CHEW birth registers missed women and these women would be the most vulnerable and least likely to get PNC?

Comment13.1- Include more detailed limitations of the study.

Response13.1: Thank you for your comment. We are included all the possible limitation of the study in the current manuscript as per your recommendation in line 12-16th of page24 of “revised manuscript with track changes”. There might be a possibility of recall bias since women were asked for events which have already happened within the past one year prior to this study.

Comment13.2- What women were missing from the study, e.g. Is it possible that the CHEW birth registers missed women and these women would be the most vulnerable and least likely to get PNC?

Response13.2: we got it as basic concern raised. The district was implementing community health information system (CHIS), in which every house hold of the district had a family folder that is updated on monthly basis. Every vital event in each kebele was recorded on timely basis since home –to- home visit has been conducted by HEWs. So, missing those women who delivered in the last 12 months from registration could be unlikely.

Comment14- What about women with stillbirths or early neonatal deaths? were they included? What difference would it make?

Response14: we are greatly appreciating your comment. Those women with still birth and early neonatal death were included in our study. Since most of contents of care provided during postpartum visits are maternal part and those mothers have to get PNC services as support tailored to address acute postpartum issues and each woman’s individual needs and it is better to make them part of the current study.

Comment15- The discussion is full of results already displayed and lacks depth in addressing the actual findings. In addition to quoting and comparing previous studies with current results, reflections on present findings should be conducted, e.g. addressing gender issues. The ideas are disclosed, however could be further elaborated in order to improve the manuscript's strength.

Response15: Thank you very much for your in-depth review and comment regarding the discussion part. We are incorporating all your comments and suggestions in each and every paragraph from the beginning to the end and all those changes are highlighted in the “revised manuscript with track changes”. In addition, some references were added in the current manuscript and those are highlighted in the revised manuscript with track changes by mentioning as’…new reference”

Comment16- PLOS authors have the option to publish the peer review history of their article (what does this mean?). If published, this will include your full peer review and any attached files. Do you want your identity to be public for this peer review? For information about this choice, including consent withdrawal, please see our Privacy Policy.

Reviewer #2: No

Response: Yes 

END________________________________________

---

## [Decision Letter · Decision Letter 1]

17 Nov 2020

PONE-D-20-05045R1

Uptake of Complete postnatal care services and its determinants among women in Ezha district, Guraghe zone, Southern Ethiopia , 2019: Community-based cross-sectional study based on the current WHO recommendation

PLOS ONE

Dear Dr. Habte,

Thank you for submitting your manuscript to PLOS ONE. After careful consideration, we feel that it has merit but does not fully meet PLOS ONE’s publication criteria as it currently stands. Therefore, we invite you to submit a revised version of the manuscript that addresses the points raised during the review process.

We look forward to receiving your revised manuscript.

Kind regards,

Onikepe Oluwadamilola Owolabi

Academic Editor

PLOS ONE

Reviewers' comments:

Reviewer's Responses to Questions

**Comments to the Author**

1. If the authors have adequately addressed your comments raised in a previous round of review and you feel that this manuscript is now acceptable for publication, you may indicate that here to bypass the “Comments to the Author” section, enter your conflict of interest statement in the “Confidential to Editor” section, and submit your "Accept" recommendation.

Reviewer #3: (No Response)

2. Is the manuscript technically sound, and do the data support the conclusions?

Reviewer #3: Partly

3. Has the statistical analysis been performed appropriately and rigorously? 

Reviewer #3: Yes

4. Have the authors made all data underlying the findings in their manuscript fully available?

Reviewer #3: Yes

5. Is the manuscript presented in an intelligible fashion and written in standard English?

Reviewer #3: No

6. Review Comments to the Author

Reviewer #3: General comment

The manuscript highlights an important issue of maternal health. However, the soundness of the methodology needs to be improved.

Major revision

Title: The title is too long; it may make confuse the reader. The year can be indicated in the methods section instead of in the title.

Methods:

- The rationale to select Ezha district as the study area should be given since there were some previous studies about the same topics in Ethiopia.

- The estimation of the study population size should be given since the study aimed to assess the coverage of PNC uptake.

- The sentence on page 6 of the sampling techniques, “The district consists of 28 Kebeles” and “Kebele is the smallest administrative unit in Ethiopia” are redundant since there is similar information under the study setting.

- Was there a sampling frame of households in each Kelebes? Please elaborate.

- The authors wrote, “A pre-tested structured questionnaire was developed by reviewing relevant literature with reasonable modifications [21-23, 36, 37].” Could you explain the rationale to use those questionnaires? Please also explain

- The theoretical framework that the researchers used for the development of the research instruments.

- Definition and operationalization of variables of the study should be placed before the data analysis.

Results:

- Figure 3, why did the authors exclude women with no PNC from the analysis? I think it should be included in the analysis.

Discussion

- The discussion is lacking the elaboration on public health/health service implications of the findings. What are the lessons from this study for other contexts of low and middle-income countries?

- The discussion is lacking on the description of the limitations and strength of the study.

Minor revision

- The manuscript contains many typos. The use of capital letters is often improper. Please do the re-editing.

- The manuscript needs an English editing

- Mean and standard deviation should be written as a number after the point (write 28.8 +5.30 instead of 28.84 + 5.306 (page 10)

- Please put the stand of each abbreviation in the table (e.g., MWR in table 3). Please also check the same issue in table 5.

- Make the figure as two dimensions one, instead of three dimensions.

- Pie charts and bar charts are not preferable for presenting the results since you already have the tables with similar content.

7. PLOS authors have the option to publish the peer review history of their article (what does this mean?). If published, this will include your full peer review and any attached files.

Reviewer #3: No

---

## [Author Response · Author response to Decision Letter 1]

25 Nov 2020

AUTHORS’ RESPONSE TO ACADEMIC EDITOR AND REVIEWERS

Dear: Onikepe Oluwadamilola Owolabi, Academic Editor

We thank you and the reviewers for a thorough reading and constructive criticism of our manuscript and for the opportunity to revise and resubmit. We are pleased to submit the revised research article “Uptake of Complete postnatal care services and its determinants among rural women in Southern Ethiopia: Community-based cross-sectional study based on the current WHO recommendation” for your consideration in the special collection of PLOS ONE. On the following pages, you will find our response to the editor’s comments. On behalf of my co-authors, I thank you for your consideration of this resubmission. We appreciate your time and look forward to your response.

Sincerely, 

Aklilu H.(MPH)(corresponding author)

aklilihabte57@gmail.com

General comments

Comment from The Review Process: Thank you for submitting your manuscript to PLOS ONE. After careful consideration, we feel that it has merit but does not fully meet PLOS ONE’s publication criteria as it currently stands. Therefore, we invite you to submit a revised version of the manuscript that addresses the points raised during the review process.

RESPONSE: Thank you for coordinating this review process. We reviewed the manuscript taking full advantage of our efforts and attempted to respond to the reviewer's comments in advance. 

Comment: Please submit your revised manuscript by Jan 01 2021 11:59 PM. If you will need more time than this to complete your revisions, please reply to this message or contact the journal office at plosone@plos.org. Response: Thank you for taking the time to submit the revised manuscript to us on time and we will attempt to submit the revised manuscript within the required timeframe.

Comment: Response: Thank you for the suggestion, but there are no changes to financial disclosure ( i.e no funding to report)

Comment: If applicable, we recommend that you deposit your laboratory protocols in protocols.io to enhance the reproducibility of your results. Protocols.io assigns your protocol its identifier (DOI) so that it can be cited independently in the future. For instructions see: http://journals.plos.org/plosone/s/submission-guidelines#loc-laboratory-protocols

Response: This search was not based on laboratories and has no laboratory protocol for depositing here.

END________________________________________

THANK YOU!!!

I. AUTHOR’S RESPONSE TO REVIEWERS’ COMMENTS

Dear: Reviewers

We thank you for a thorough reading and constructive criticism of our manuscript and for the opportunity to revise and resubmit. We are pleased to submit the improved research article “Uptake of Complete postnatal care services and its determinants among rural women in Southern Ethiopia: Community-based cross-sectional study based on the current WHO recommendation” for your consideration in the special collection of PLOS ONE. On the following pages, you will find our response to the reviewers’ comments. On behalf of my co-authors, I thank you for your consideration of this resubmission. We appreciate your time and look forward to your response independently as for reviewer#3.

Sincerely, 

Aklilu H. (MPH)(corresponding author)

aklilihabte57@gmail.com

AUTHOR’S RESPONSE TO REVIEWER#3

Comments to the Author

Comment1: If the authors have adequately addressed your comments raised in a previous round of review and you feel that this manuscript is now acceptable for publication, you may indicate that here to bypass the “Comments to the Author” section, enter your conflict of interest statement in the “Confidential to Editor” section, and submit your "Accept" recommendation.

Reviewer #3: (No Response)

Response: First, we would like to hear from you and your suggestion. We have tried to address all of the potential comments and suggestions raised by reviewers 1 and 2 in the previous review process and now we are also encouraged to consider all of your suggestions and comments. 

Comment2: Is the manuscript technically sound, and do the data support the conclusions?

Reviewer #3: Partly

Response: We appreciate what you have to say. After your comments and suggestions, we have emphasized making the manuscript technically sound from its writing to the detailed methodology and analysis portion. To ensure representativeness, the sample size was determined for the first and second objectives and the maximum sample size was used for the study. Concerning the conclusion, it is based on the findings of the study without ignoring our initial objectives.

Comment3: Has the statistical analysis been performed appropriately and rigorously?

Reviewer #3: Yes

Response: We thank you for your constructive and positive viewpoint regarding the analysis portion of the study.

Comment4: Have the authors made all data underlying the findings in their manuscript fully available?

The PLOS Data policy requires authors to make all data underlying the findings described in their manuscript fully available without restriction, with rare exceptions (please refer to the Data Availability Statement in the manuscript PDF file). The data should be provided as part of the manuscript or its supporting information, or deposited in a public repository. For example, in addition to summary statistics, the data points behind means, medians, and variance measures should be available. If there are restrictions on publicly sharing data—e.g. participant privacy or use of data from a third party—those must be specified.

Reviewer #3: Yes

Response: We appreciate your assurance regarding the availability of all necessary data.

Comment5: Is the manuscript presented in an intelligible fashion and written in standard English?

Reviewer #3: No

Response: We greatly appreciate your efforts to thoroughly review the manuscript to detect typographical or grammatical errors. In the present revision of the manuscript, we have placed a strong emphasis on this part and attempt to correct all possible grammatical and typological errors in detail. All necessary changes and corrections in each section of the manuscript have been indicated in the “revised manuscript with track changes” file.

6. Review Comments to the Author

General comment of reviewer 3: The manuscript highlights an important issue of maternal health. However, the soundness of the methodology needs to be improved.

Response: First of all, we would appreciate your constructive input. Regarding the soundness of the methodology, we are trying to explore the whole methodology section of the manuscript after your comment and we will try to make it sounder. All possible revisions in the “Data Collection Tools and Personnel”, “Data Quality”, and “Data Analysis” sections were highlighted in the “Revised Manuscript with Change Tracking” file.

Major revision

Comment1: The title is too long; it may make confuse the reader. The year can be indicated in the methods section instead of in the title.

Response: Thank you for your pertinent comment to reduce the confusion concerning the title and now it is corrected as “Uptake of Complete postnatal care services and its determinants among rural women in Southern Ethiopia: Community-based cross-sectional study based on current WHO recommendation”

Comments in the Method part

Comment: The estimation of the study population size should be given since the study aimed to assess the coverage of PNC uptake.

Response: Thank you for your concern about determining the sample size. We calculate the sample size in two ways; for the first and second objectives, as we try to clarify that in the 1st to 10th line of the “sample size determination” portion of the “Manuscript file”. Subsequently, the maximum sample size obtained from the second objective (i.e. for the CPNC determinants) was used for the study. All of this was done according to our advisors during the proposal development phase.

Comment: The sentence on page 6 of the sampling techniques, “The district consists of 28 Kebeles” and “Kebele is the smallest administrative unit in Ethiopia” are redundant since there is similar information under the study setting.

Response: thank you for an in-depth view and now it is omitted and highlighted in the “sampling technique” section of the “revised manuscript with track changes” file

Comment: Was there a sampling frame of households in each Kebeles? Please elaborate.

Response: Firstly, we would like you to ask us to respond to this sampling question as part of the research process. Initially, we obtained the list of women who gave birth within the last 12 months and who remained for six weeks or more after delivery from the registry of each health post, because each delivery was recorded daily. After verifying the presence of these eligible women, codes/numbers were provided for the homes, and a sampling frame was developed for each Kebele. All of these procedures were highlighted in the “Sampling Technique” section of the “Revised manuscript with Track Changes” file, lines 6-11.

Comment: The authors wrote, “A pre-tested structured questionnaire was developed by reviewing relevant literature with reasonable modifications [21-23, 36, 37].” Could you explain the rationale to use those questionnaires? Please also explain.

Response: Just to show that the tool/questionnaire we used to collect the data was developed by looking at different pieces of literature in the area of interest. The references mentioned have already been the subject of postnatal care research. The word “Pre-tested” is indicated to show that the questionnaire was tested prior to the start of the research and is well elaborated in the “Data Quality Management” section of the “Revised Manuscript with Track Changes”, page8, lines 5-8.

Comment: The theoretical framework that the researchers used for the development of the research instruments.

Response: We appreciate the question. As we attempt to respond to the above comment, we developed the research instrument after reviewing various research carried out in postnatal care and more specifically on complete postnatal care.

Comment: Definition and operationalization of variables of the study should be placed before the data analysis.

Response: We accept your recommendation and do so accordingly.it is highlighted in the ‘revised manuscript with track change’, Page 8.

 Comments in the result part

Comment: Figure 3, why did the authors exclude women with no PNC from the analysis? I think it should be included in the analysis.

Response: We have taken this as a mistake and we are doing so based on your recommendation. We also omitted this figure in the current revised manuscript since it is already mentioned in the form of a declaration.

 Comment to the discussion section.

Comment: The discussion is lacking the elaboration on public health/health service implications of the findings. What are the lessons from this study for other contexts of low and middle-income countries?

Response: Thank you very much for this comment, and we have received these suggestions and recommendations as a necessary contribution to strengthen the implications of our findings. We try to underline what the results mean and what measures should be taken by the stakeholders based on the findings and all these are mentioned and highlighted in the 'discussion' part of the 'manuscript with track changes' file.

Comment: The discussion is lacking in the description of the limitations and strength of the study.

Response: thank you for the comment. We understood that it was our fault, in a manuscript submitted earlier, to clearly show the strength and limitations of our study. But now we have tried to clarify these limitations and strengths accordingly and have highlighted them in the last paragraph of the 'discussion portion' of the 'revised manuscript with track changes' file, Page22

Minor revision

Comment: The manuscript contains many typos. The use of capital letters is often improper. Please do the re-editing.

Response: We would like to appreciate you for an in-depth examination of our manuscript. We revised the whole document according to your recommendations, line by line, and we tried to manage these problems. We tried to show all of these changes in the “revised manuscript with track changes”. Comment: Mean and standard deviation should be written as a number after the point (write 28.8 +5.30 instead of 28.84 + 5.306 (page 10)

Response: this error occurred in the 'Socio-demographic Characteristics of Respondents' portion of the 'result' part and is now corrected.

Comment: Make the figure as two dimensions one, instead of three dimensions.

Response: Thank you very much for your meticulous observation to our work. But initially, we convert these figures using Preflight Analysis Conversion Engine (PACE) per the PLOS ONE manuscript submission guideline. If other options are available, we are welcome to do so.

Comment: Pie charts and bar charts are not preferable for presenting the results since you already have the tables with similar content.

Response: thank you for your valuable comment. According to your suggestion, we have omitted the figures which lead us to redundancy.

Comment: PLOS authors have the option to publish the peer review history of their article (what does this mean?). If published, this will include your full peer review and any attached files. Response: No 

Comment: Do you want your identity to be public for this peer review? For information about this choice, including consent withdrawal, please see our Privacy Policy.

Reviewer #3: No

Response: no 

END________________________________________

With kind regards!!

---

## [Decision Letter · Decision Letter 2]

13 Jan 2021

PONE-D-20-05045R2

Uptake of Complete postnatal care services and its determinants among rural women in  Southern Ethiopia : Community-based cross-sectional study based on the current WHO recommendation

PLOS ONE

Dear Dr. Habte,

Thank you for submitting your manuscript to PLOS ONE. After careful consideration, we feel that it has merit but does not fully meet PLOS ONE’s publication criteria as it currently stands. Therefore, we invite you to submit a revised version of the manuscript that addresses the points raised during the review process.

We look forward to receiving your revised manuscript.

Kind regards,

Frank T. Spradley

Academic Editor

PLOS ONE

Reviewers' comments:

Reviewer's Responses to Questions

**Comments to the Author**

1. If the authors have adequately addressed your comments raised in a previous round of review and you feel that this manuscript is now acceptable for publication, you may indicate that here to bypass the “Comments to the Author” section, enter your conflict of interest statement in the “Confidential to Editor” section, and submit your "Accept" recommendation.

Reviewer #3: All comments have been addressed

2. Is the manuscript technically sound, and do the data support the conclusions?

Reviewer #3: Yes

3. Has the statistical analysis been performed appropriately and rigorously? 

Reviewer #3: Yes

4. Have the authors made all data underlying the findings in their manuscript fully available?

Reviewer #3: No

5. Is the manuscript presented in an intelligible fashion and written in standard English?

Reviewer #3: Yes

6. Review Comments to the Author

Reviewer #3: There are typo, such as the use of capital letter. The writing of p=0.000 should be p<0.001. Please check and revise.

7. PLOS authors have the option to publish the peer review history of their article (what does this mean?). If published, this will include your full peer review and any attached files.

Reviewer #3: No

---

## [Author Response · Author response to Decision Letter 2]

13 Jan 2021

AUTHORS’ RESPONSE TO ACADEMIC EDITOR AND REVIEWERS

Dear: Frank T. Spradley, Academic Editor

We thank you and the reviewers for a thorough reading and constructive criticism of our manuscript and for the opportunity to revise and resubmit. We are pleased to submit the revised research article “Uptake of Complete postnatal care services and its determinants among rural women in Southern Ethiopia: Community-based cross-sectional study based on the current WHO recommendation” for your consideration in the special collection of PLOS ONE. On the following pages, you will find our response to the editor’s comments. On behalf of my co-authors, I thank you for your consideration of this resubmission. We appreciate your time and look forward to your response.

Sincerely, 

Aklilu H.(MPH)(corresponding author)

Comments From Academic Editor

Thank you for submitting your manuscript to PLOS ONE. After careful consideration, we feel that it has merit but does not fully meet PLOS ONE’s publication criteria as it currently stands. Therefore, we invite you to submit a revised version of the manuscript that addresses the points raised during the review process.

We look forward to receiving your revised manuscript.

Kind regards,

Frank T. Spradley

Academic Editor

Response: Thank you for your kind feedback and, as per your recommendation, the revised manuscript has been submitted here.

END________________________________________

THANK YOU!!!

I. AUTHOR’S RESPONSE TO REVIEWERS’ COMMENTS

Dear: Reviewer

We thank you for a thorough reading and constructive criticism of our manuscript and for the opportunity to revise and resubmit. We are pleased to submit the improved research article “Uptake of Complete postnatal care services and its determinants among rural women in Southern Ethiopia: Community-based cross-sectional study based on the current WHO recommendation” for your consideration in the special collection of PLOS ONE. On the following pages, you will find our response to the reviewers’ comments. On behalf of my co-authors, I thank you for your consideration of this resubmission. We appreciate your time and look forward to your response independently as for reviewer#3.

Sincerely, 

Aklilu H. (MPH)(corresponding author)

aklilihabte57@gmail.com

Comments to the Author

1. If the authors have adequately addressed your comments raised in a previous round of review and you feel that this manuscript is now acceptable for publication, you may indicate that here to bypass the “Comments to the Author” section, enter your conflict of interest statement in the “Confidential to Editor” section, and submit your "Accept" recommendation.

Reviewer #3: All comments have been addressed

 Response: Thank you very much

2. Is the manuscript technically sound, and do the data support the conclusions?

Reviewer #3: Yes

 Response: We are pleased with your trust that our manuscript is technically sound.

3. Has the statistical analysis been performed appropriately and rigorously?

Reviewer #3: Yes________________________________________

 Response: Thank you so much for agreeing that our analysis is appropriate and rigorous.

4. Have the authors made all data underlying the findings in their manuscript fully available?

The PLOS Data policy requires authors to make all data underlying the findings described in their manuscript fully available without restriction, with rare exceptions (please refer to the Data Availability Statement in the manuscript PDF file). The data should be provided as part of the manuscript or its supporting information, or deposited to a public repository. For example, in addition to summary statistics, the data points behind means, medians and variance measures should be available. If there are restrictions on publicly sharing data—e.g. participant privacy or use of data from a third party—those must be specified.

Reviewer #3: No

 Response: Thank you for your reminder and we have now listed these data in the "supportive information" section of the revised manuscript, on the last page.

5. Is the manuscript presented in an intelligible fashion and written in standard English?

Reviewer #3: Yes

6. Review Comments to the Author

Please use the space provided to explain your answers to the questions above. You may also include additional comments for the author, including concerns about dual publication, research ethics, or publication ethics. (Please upload your review as an attachment if it exceeds 20,000 characters 

Reviewer #3: There are typo, such as the use of capital letter. The writing of p=0.000 should be p<0.001. Please check and revise.

Response: Thanks for the comment, and now all the above typo errors have been corrected and are highlighted in the “revised manuscript with track changes”

7. PLOS authors have the option to publish the peer review history of their article (what does this mean?). If published, this will include your full peer review and any attached files.

Do you want your identity to be public for this peer review? For information about this choice, including consent withdrawal, please see our Privacy Policy.

Reviewer #3: No

Response: Yes

Response: Thank you for your reminder about the figure format and after confirmation, we checked and submitted them.

---

## [Editor Report · Decision Letter 3]

15 Jan 2021

PONE-D-20-05045R3

Uptake of Complete postnatal care services and its determinants among rural women in  Southern Ethiopia : Community-based cross-sectional study based on the current WHO recommendation

PLOS ONE

Dear Dr. Habte,

Thank you for submitting your manuscript to PLOS ONE. After careful consideration, we feel that it has merit but does not fully meet PLOS ONE’s publication criteria as it currently stands. Therefore, we invite you to submit a revised version of the manuscript that addresses the points raised during the review process.

The manuscript has been revised according to the reviewers' comments. However, one noticeable typographical error remains: in table 4, it should read "body weight" not "bodyweigh". Please correct.

We look forward to receiving your revised manuscript.

Kind regards,

Frank T. Spradley

Academic Editor

PLOS ONE

---

## [Author Response · Author response to Decision Letter 3]

15 Jan 2021

AUTHORS’ RESPONSE TO ACADEMIC EDITOR 

Dear: Frank T. Spradley, Academic Editor

We thank you and the reviewers for a thorough reading and constructive criticism of our manuscript and for the opportunity to revise and resubmit. We are pleased to submit the revised research article “Uptake of Complete postnatal care services and its determinants among rural women in Southern Ethiopia: Community-based cross-sectional study based on the current WHO recommendation” for your consideration in the special collection of PLOS ONE. On the following pages, you will find our response to the editor’s comments. On behalf of my co-authors, I thank you for your consideration of this resubmission. We appreciate your time and look forward to your response.

Sincerely, 

Aklilu H.(MPH)(corresponding author)

Comments From Academic Editor

PONE-D-20-05045R3

Uptake of Complete postnatal care services and its determinants among rural women in Southern Ethiopia : Community-based cross-sectional study based on the current WHO recommendation

PLOS ONE

Dear Dr. Habte,

Thank you for submitting your manuscript to PLOS ONE. After careful consideration, we feel that it has merit but does not fully meet PLOS ONE’s publication criteria as it currently stands. Therefore, we invite you to submit a revised version of the manuscript that addresses the points raised during the review process.

Comment: The manuscript has been revised according to the reviewers' comments. However, one noticeable typographical error remains: in table 4, it should read "body weight" not "bodyweigh". Please correct.

Response: We appreciate your meticulous review of our work. In the revised manuscript, we corrected it and highlighted it in Table 4, page 14 of the "Revised Manuscript With Track Changes"

We look forward to receiving your revised manuscript.

Kind regards,

Frank T. Spradley

Academic Editor

PLOS ONE

END________________________________________

THANK YOU!!!

---

## [Editor Report · Decision Letter 4]

18 Jan 2021

Uptake of Complete postnatal care services and its determinants among rural women in  Southern Ethiopia : Community-based cross-sectional study based on the current WHO recommendation

PONE-D-20-05045R4

Dear Dr. Habte,

We’re pleased to inform you that your manuscript has been judged scientifically suitable for publication and will be formally accepted for publication once it meets all outstanding technical requirements.

Kind regards,

Frank T. Spradley

Academic Editor

PLOS ONE

---

## [Editor Report · Acceptance letter]

22 Jan 2021

PONE-D-20-05045R4 

Uptake of Complete postnatal care services and its determinants among rural women in Southern Ethiopia: Community-based cross-sectional study based on the current WHO recommendation 

Dear Dr. Habte:

I'm pleased to inform you that your manuscript has been deemed suitable for publication in PLOS ONE. Congratulations! Your manuscript is now with our production department. 

Kind regards, 

on behalf of

Dr. Frank T. Spradley 

Academic Editor

PLOS ONE